# Low-Power Failure Detection for Environmental Monitoring Based on IoT

**DOI:** 10.3390/s21196489

**Published:** 2021-09-28

**Authors:** Jiaxi Liu, Weizhong Gao, Jian Dong, Na Wu, Fei Ding

**Affiliations:** 1Jiangsu Key Laboratory of Broadband Wireless Communication and Internet of Things, Nanjing University of Posts and Telecommunications, Nanjing 210003, China; liujiaxi@njupt.edu.cn (J.L.); wuna@njupt.edu.cn (N.W.); 2School of Internet of Things, Nanjing University of Posts and Telecommunication, Nanjing 210003, China; 3State Key Laboratory for Novel Software Technology, Nanjing University, Nanjing 210023, China; scott.gao@oracle.com; 4School of Computer Science and Technology, Harbin Institute of Technology, Harbin 150001, China; dj@ftcl.hit.edu.cn

**Keywords:** failure detection, IoT, environmental monitoring, battery consumption, Weibull distribution

## Abstract

Many environmental monitoring applications that are based on the Internet of Things (IoT) require robust and available systems. These systems must be able to tolerate the hardware or software failure of nodes and communication failure between nodes. However, node failure is inevitable due to environmental and human factors, and battery depletion in particular is a major contributor to node failure. The existing failure detection algorithms seldom consider the problem of node battery consumption. In order to rectify this, we propose a low-power failure detector (LP-FD) that can provide an acceptable failure detection service and can save on the battery consumption of nodes. From simulation experiments, results show that the LP-FD can provide better detection speed, accuracy, overhead and battery consumption than other failure detection algorithms.

## 1. Introduction

The Internet of Things (IoT) has been gaining momentum in both the industry and research communities due to an explosion in the number of smart mobile devices and sensors and the potential applications of the data produced from a wide spectrum of domains [1,2]. Among the IoT application domains, environmental monitoring is receiving increased attention as environmental technology becomes a key area of global sustainable development. For example, underwater resource management [3], wetland monitoring systems [4], emergency management communities [5], urban public safety emergency management early warning systems [6], and so on. These applications require the IoT to maintain a high availability for reliable execution. However, failure is inevitable due to various environmental factors and sensor hardware or software malfunctions; in particular, the inability of sensors to recharge batteries. Thus, it is a challenge to maintain the high availability of environmental monitoring IoT applications.

Failure detection is an essential component of building highly available systems, especially if there are safety applications in the system [7]. Failure detection can periodically identify the state of neighbor nodes, then output the results to achieve routing discovery, application deployment, real time communication, etc. Thus, the existence of failure detection can ensure the high availability of IoT applications. An effective failure detection algorithm can find failure nodes accurately and promptly so that the behavior of the system can be adjusted as soon as possible. At present, many studies regarding failure detection algorithms based on a heartbeat protocol are proposed for distribution systems [8,9,10,11,12]. However, these failure detection algorithms do not consider the application environment of the IoT. For example, a large number of sensors in IoT applications do not have strong computing capabilities and are lacking a sufficient power supply due to the special application environment. Therefore, these failure detection algorithms are not adequate for the IoT.

In this paper, we focus on the problem of failure detection when remote nodes do not have a sufficient power supply [13]. Accordingly, our failure detection algorithm does not consume a large amount of node power and can mitigate the problem of sensor energy consumption to achieve environmental monitoring in remote areas. To facilitate environmental monitoring in remote or inaccessible areas without a sufficient power supply, we present a low-power failure detector (LP-FD) for IoT applications. A key design aspect of the LP-FD is to employ a variable detection period. We assume that the online timing of sensors follows the Weibull distribution, thus the detection period of the LP-FD can be calculated by the reliability function of the Weibull distribution. When detection begins, the detection period of the LP-FD is set to be longer due to the high reliability of the sensors. While in subsequent detections, the detection period of the LP-FD is set to be shorter due to the low reliability of the sensors. Compared to traditional FDs, the LP-FD needs fewer heartbeat messages to achieve failure detection. Thus, the LP-FD can save on communication overhead in order to reduce sensor battery consumption. The main contributions of this paper are presented as follows:We have designed a novel FD for environmental monitoring based on the IoT that ensures a high availability, and a reliable execution, of applications.The detection period can be calculated by the reliability function of the Weibull distribution, and it has a proportional relationship to the reliability of the sensors.Due to the variable detection period, the number of communications per unit time is reduced, which saves on sensor power consumption and detection overhead.

The rest of this paper is organized as follows. In Section 2, related work regarding the environmental monitoring of the IoT and failure detection is introduced. Section 3 introduces the system model. The implementation of the LP-FD is proposed in Section 4. The simulation results are reported in Section 5. Finally, the work is concluded in Section 6.

## 2. Related Work

### 2.1. Environmental Monitoring of IoT

The increasingly serious issue of environmental pollution has promoted the rapid development of environmental monitoring [14]. Environmental monitoring has been conducted for more than 50 years. At present, IoT technology is being applied in the field of environmental monitoring as a new technology [15]. In addition, it has brought new opportunities for many technologies, such as intelligent sensing environmental monitoring technology, embedded technology, and so on. 

In many countries, intelligent environmental monitoring has become very popular. Many systems use various wireless LAN protocols to achieve environmental monitoring, such as the Home Radio Frequency (Home-RF), which is used in the sensor networks of some home devices, and the ZigBee protocol, with the physical layer and the medium access control layer following the IEEE 802.15.4 standard [16,17,18]. An ecological monitoring system for the distribution and habits of toads has been developed by the Australian Government [19]. The seabirds of Big Duck Island are monitored by an ecological monitoring system [20]. The IoT technology has also been widely used in the field of environmental monitoring. It uses monitoring devices, rather than sensing devices, and connects terminal testing devices or connects with end customers, environmental protection departments, and personal digital display monitoring systems, allowing people to understand the environmental conditions more intuitively and quickly.

There are three levels in environmental monitoring techniques based on the IoT. The first level is the intelligent sensing layer, the second level is the network communication layer, and the third level is the application layer (as shown in Figure 1).

The perception layer contains various sensors, and the systems on the sensors are used to obtain the environmental parameters. 

The network layer is mainly used to transmit data by 5G, GPRS, and ZigBee [21]. The users can conveniently use these data with a terminal computer or mobile.

The application layer is mainly used to analyze and process the information and data, to make reasonable controls and decisions, and to realize intelligent management, application and service.

### 2.2. Failure Detection

With the development of distributed systems, failure detection technology has been an important part of building a highly available distributed system. This technology has received a lot of attention since its emergence, and many different types of failure detectors have been proposed, such as the Cassandra distributed database, which uses an accrual failure detector to detect node failure [22]. Aiming at fault tolerance distributed systems, Chandra and Toueg proposed the concept of failure detection for the first time. At the same time, they defined two properties (completeness and accuracy) to describe the detection capability of a failure detector. “Completeness” is the ability of a failure detector to eventually find the node failure. “Accuracy” is the ability of a failure detector to avoid false detection.

Many failure detectors that are implemented employ the heartbeat protocol or the ping protocol. The heartbeat protocol is where the monitored nodes periodically send heartbeat messages to a failure detector, then the failure detector determines the state of the nodes according to whether it receives the heartbeat messages. Whereas the ping protocol is where a failure detector actively sends query messages to the monitored nodes, then the failure detector determines the state of the nodes according to the response of the monitored nodes. There are some other important failure detectors that work as follows.

Chen et al. [23] proposed a Quality of Service based (QoS-based) failure detector in accordance with the probability network model. In the failure detector, a node *p* sends a heartbeat message *m* to a node *q* every unit of time. A sliding window located at node *q* can be used to store the last *n* heartbeat messages $m_{1},m_{2},\ldots ,m_{n}$. $A_{1},A_{2},\ldots ,A_{n}$ are the receipt times according to *q*’s local clock. Subsequently, the expected arrival time of the next heartbeat message is estimated by:(1)$$EA_{\left(k+1\right)}=\frac{1}{n}\sum_{i=k-n-1}^{k}(A_{i}-\eta \times i)+(k+1)\eta$$
where η is the sending interval, decided by the QoS requirement of the user. In the failure detector, the concept of freshpoint is introduced. The freshpoint is the timeout threshold used to determine whether the monitored node has failed. The freshpoint $\tau_{k+1}$ of the next heartbeat message consists of $EA_{k+1}$ and the constant safety margin SM. One has:(2)$$\tau_{k+1}=EA_{k+1}+SM$$
where *SM* means that an additional amount of time is added to the timeout value to improve the detection accuracy. The arrival time of the next heartbeat message is estimated by the constant safety margin in this failure detector.

Based on Chen’s FD, Tomsic et al. [8] proposed a two sliding windows failure detector (2W-FD) that can adapt to sudden changes in unstable network scenarios. The sliding window is a space used to store the arrival time of the heartbeat messages. In the 2W-FD, there are two sliding windows for storing the past received messages; a small one is used to store a few received messages, and a bigger one is used to store a large amount of received messages. The small window can cope with abrupt changes in network conditions, while the bigger window deals better with stable or slowly changing conditions. The 2W-FD is able to compute two expected arrival times, $EA_{l+1}^{n_{1}}$ and $EA_{l+1}^{n_{2}}$, according to the two sliding windows. Finally, the bigger estimation is used to compute the next freshness point $\tau_{l+1}$:(3)$$\tau_{l+1}=\max(EA_{l+1}^{n_{1}},EA_{l+1}^{n_{2}})+SM$$
where *SM* is a constant safety margin.

A continuous value φ is used to represent the suspicion level of the monitored node in φ-FD [9]. This method is different from a binary method, which uses trusted or suspect as the output. In the implementation of φ-FD, a sliding window is used to store the most recent arrival time of the heartbeat messages. It is supposed that the arrival time of the heartbeat messages follows a normal distribution. Subsequently, the value of φ can be calculated as follows:(4)$$\varphi (T_{now})=-log_{10}(P_{later}(T_{now}-T_{last}))$$
where $T_{last}$ is the time when the fresh heartbeat message arrives, $T_{now}$ is the current time, and $P_{later}(t)$ is the probability that the arrival time of the fresh heartbeat message is more than t time units later than the previous one. Based on the assumption of normal distribution, $P_{later}(t)$ can be computed as follows:(5)$$P_{later}(t)=\frac{1}{\sigma \sqrt{2\pi }}\int_{t}^{\infty }e^{-\frac{{\left(x-\mu \right)}^{2}}{2\sigma^{2}}}dx=1-F(t)$$
where F(t) is the cumulative distribution function of a normal distribution with mean μ and variance $\sigma^{2}$. φ-FD can provide a value of φ to the applications that query the φ-FD at time $T_{now}$. Subsequently, each application can carry out different actions according to its threshold Φ, which is set by different QoS requirements. Thus, the different QoS requirements of multiple applications can be met simultaneously.

The ED-FD [24], which is based on exponential distribution, is similar to the φ-FD. In the ED-FD, it is assumed that the arrival time of the heartbeat messages follows exponential distribution. Thus, the suspicion level of the monitored node, $e_{d}$, can be calculated as follows:(6)$$e_{d}=F(T_{now}-T_{last})$$
(7)$$F(t)=1-e^{-\frac{t}{\mu }}$$
where $T_{now}$, $T_{last}$, and μ have the same meaning as for the φ-FD. For the ED-FD, the threshold is $E_{d}$.

### 2.3. QoS Metrics of Failure Detection

For some distributed applications, there are some timing constraints on the behaviors of failure detectors. A failure detector cannot meet the requirements of these applications if a node starts to be suspected long after it fails, or if the failure detector makes too many mistakes. In order to solve this problem, Chen proposed a series of metrics to restrain the behavior of failure detectors. These metrics can explain how quickly node failure is found and how much error detection is avoided. Moreover, they can describe the performance of a failure detector quantitatively. In these metrics, *T* represents that a node works normally and *S* represents that a node is suspected of failure. When a *T*-transition occurs, it means that the failure detector corrects a false suspicion. When an *S*-transition occurs, it means that the failure detector suspects a node failure. Based on the above description, following are some primary metrics to describe the QoS of a failure detector:Detection time ($T_{D}$) is from the moment a node crashes to the moment it is permanently suspected, i.e., when the final *S*-transition occurs.Mistake rate ($\lambda_{M}$) is the number of false suspicions a failure detector makes per unit time, i.e., it is used to describe the frequency of false suspicions of a failure detector.Query accuracy probability ($Q_{A}$) is the probability that the output of a failure detector is correct at a random time.

The first metric is used to describe the detection speed of a failure detector; the others are used to describe the detection accuracy of a failure detector. Because the mistake rate is not sufficient to describe the detection accuracy of a failure detector, it also employs query accuracy probability to indicate the detection accuracy. For example, node *p* is detected by FD_1_ and FD_2_ in Figure 2. In the whole detection process (16 s), node *p* is in a normal state. In Figure 2, *T* represents that the output of the failure detector is trusted, while the *S* represents that the output of the failure detector is suspect. For FD_1_, there are two false suspicions in the whole detection process. According to the definition of mistake rate, the mistake rate of FD_1_ is 2/16=0.125. The output of trust lasts 12 s, and it accounts for 12/16=0.75 of the overall output. Therefore, this means that the query accuracy probability of FD_1_ is 0.75. For FD_2_, there are two false suspicions in the whole detection process. According to the definition of mistake rate, the mistake rate of FD_2_ is 2/16=0.125. The output of trust lasts 8 s, and it accounts for 8/16=0.5 of the overall output. Thus, this means that the query accuracy probability of FD_2_ is 0.5. Both failure detectors have the same mistake rate (0.125), but they have different query accuracy probabilities (0.75 and 0.5).

Detection overhead ($O_{D}$) is the traffic used to find a failure node. It can be measured by recording the average number of messages for the detection purpose.

## 3. Proposed System

### 3.1. Network Model

The network model is the basic factor that must be considered in the design of a failure detector. It records the state of the monitored nodes for the suspect list in each failure detector. When a node is suspected by any failure detector, the failure detector must transmit this information to other nodes in network. However, it is very time consuming and load consuming to let all the nodes know this failure information in such a large-scale system. In this paper, we consider the concept that each failure detector only connects to partial nodes and is responsible for detecting them. More specifically, each failure detector is responsible for detecting 1-hop neighbor nodes. Failure information can be transmitted along neighbor nodes. 

### 3.2. Link Failure

In the IoT, wireless communication channels are unstable. Radio interference is a main factor of link failure. If link failure occurs, the packets will be lost. In most cases, a failure detector can correct its own false suspicions because the link failure is temporary. In this paper, we consider that communication channels are unreliable. We assume that the communication channel is a fair lossy channel [25]. This channel allows packet loss, but it cannot copy or modify the message and create a new message. Additionally, node *q* can receive message *m* if node *p* continuously sends that message.

### 3.3. Node Failure

In a malicious environment, sensors may have antenna failure, circuit failure, battery leakage, and other problems. These problems will lead to sensor failure and will affect the system performance. For sensor failure, we consider it belongs to a crash-stop. When a sensor has a crash-stop, it cannot send or receive messages. Under normal circumstances, a sensor will always send or receive messages without failure. Sensor *p* can determine whether its neighbor sensor *q* is normal according to the information in the received message.

## 4. Implementation of Low-Power Failure Detector

### 4.1. The Detection Period

In a failure detector, the detection performance is seriously affected by the detection period. For example, a longer detection period will increase the detection time and reduce the detection accuracy, whereas a shorter detection period will generate more heartbeat messages and increase the detection overhead, which means that more communication cost and computation cost will be consumed. In the IoT, it is normal for self-powered sensors to fail due to battery exhaustion. Excessive detection overhead will accelerate battery consumption and cause sensor failure. Thus, we need to find a reasonable detection period configuration method to balance the detection time, detection accuracy, and detection overhead. In this paper, we propose a new method for determining the detection period in the IoT (as shown in Algorithm 1). The definition of the parameters involved in this method are shown in Table 1.

**Algorithm 1.** Detection Period.**Input:** $R_{req}$, $\eta_{\min}$, $T_{now}$, α,β**Output:** η1.    $R(T_{now})=e^{-{\left(T_{now}/\alpha \right)}^{\beta }}$
2.    **if** $R(T_{now})>R_{req}$
3.      **then** $t_{req}=\alpha \cdot {\left(-ln^{R_{req}}\right)}^{1/\beta }$;4.        $\Delta t=t_{req}-T_{now}$;5.        **if** $\Delta t>n\cdot \eta_{\min}$ (*n*>1)6.          **then** $\eta =n\cdot \eta_{\min}$;7.          **else if** $\eta_{\min}<\Delta t<n\cdot \eta_{\min}$
8.            **then** η=Δt;9.            **else**10.              $\eta =\eta_{\min}$;11.            **end if**12.         **end if**13.      **else**14.        $\eta =\eta_{\min}$;15.   **end if**

Considering the general failure of sensors and the exhaustion of sensor batteries, we assume that the reliability of a sensor follows the Weibull distribution [26]. Therefore, the reliability of a sensor over time can be described by:(8)$$R(t)=e^{-{\left(t/\alpha \right)}^{\beta }}$$
where the parameters α and β are used to adjust the reliability function.

According to the reliability function, the reliability value $R(t_{i})$ of a sensor can be calculated at a certain time, $t_{i}$. If this reliability value $R(t_{i})$ is greater than the preset reliability value $R_{req}$, we can calculate the detection period η (as shown in Algorithm 1).

By transforming the reliability function, we can attain:(9)$$t=\alpha \cdot {\left[-\ln^{R(t)}\right]}^{1/\beta }$$

We can obtain a time value $t_{req}$ by introducing the preset reliability value $R_{req}$ into Equation (9). Subsequently, we can attain a time difference:(10)$$\Delta t=t_{req}-t_{i}$$

If this time difference $\Delta t>n\cdot \eta_{\min}$, we use $n\cdot \eta_{\min}$ as the detection period to ensure detection accuracy (lines 5 and 6). If $\eta_{\min}<\Delta t<n\cdot \eta_{\min}$, we use Δt as the detection period (lines 7 and 8). Otherwise, we use $\eta_{\min}$ as the detection period (lines 9 to 14). Every time a heartbeat message is sent, we re-calculate the detection period.

### 4.2. Implementation of Low-Power Failure Detector

In environmental monitoring based on the IoT, there are many sensors used to monitor environment and transmit data (as shown in Figure 3). In such a large-scale system, sensor failure caused by software and hardware failure becomes inevitable. Thus, the system needs to know the status of sensors in a timely fashion to ensure the implementation of applications. For example, when a sensor fails (the red node is the failed node), all data transmitted through this node will not reach the destination. This means that the old path through the failed sensor is useless. If the system does not know how many such failed nodes exist, its availability will be greatly reduced. The purpose of a failure detector is to find the failed sensor in the system in time. By employing a failure detector, the system can find the failed sensor and then remove it from the system topology. Finally, the system builds a new path to transmit data using normal sensors. In the IoT, apart from sensor hardware and software failure, battery depletion is also an important failure factor in sensor failure.

To reduce the impact of failure detection on sensor battery consumption, an LP-FD is proposed. When the receiver obtains a heartbeat message, the message delay $d_{i}$ can be calculated by:(11)$$d_{i}=T_{now}-T_{pre}$$
where $T_{pre}$ is the arrival time of the previous heartbeat message and $T_{now}$ is the arrival time of the new heartbeat message. If a message is lost, it is difficult to measure the communication delay between the sender and the receiver. In light of the impact of message loss, our approach uses the average method to deal with the problem. In detail, we can recompute the value of the delay by:(12)$$d_{i}=\left(T_{now}-T_{pre}\right)/N_{l}$$
where $N_{l}$ is the number of lost heartbeat messages.

It is assumed that the value of $d_{i}$ is equal to the message delay of the next heartbeat message $d_{i+1}$. Thus, the expected arrival time of the next heartbeat message can be calculated by:(13)$$EA_{i+1}=\sum_{k=0}^{i}ID_{k}\cdot \eta_{k}+d_{i}$$
where $ID_{k}$ is the sequence number of the heartbeat message and $\eta_{k}$ is the *k*-th detection period.

Based on the single exponential smoothing method, we can calculate the predictive delay ${\hat{d}}_{i+1}$ as follows:(14)$${\hat{d}}_{i+1}=k*{\hat{d}}_{i}+(1-k)*d_{i}$$
where k(0≤k≤1) is a constant between 0 and 1, which controls how rapidly the ${\hat{d}}_{i+1}$ adapts to the delay change. Therefore, the safety margin (*SM*) can be estimated by:(15)$$SM_{i+1}=\varepsilon \left|{\hat{d}}_{i+1}-d_{i}\right|$$
where ε is a variable, chosen so that there is an acceptably small probability that the delay for the heartbeat message will exceed the timeout. Finally, we can compute the freshpoint for heartbeat message (i+1) by:(16)$$\tau_{i+1}=EA_{i+1}+SM_{i+1}$$

In an LP-FD, the heartbeat approach is used as the basic failure detection strategy. To simplify the description, it is supposed that there are two sensors, *p* and *q*, in the system. Sensor *q* is responsible for detecting sensor *p*. Algorithm 2 shows the detailed detection algorithm.

**Algorithm 2.** Low-power Failure Detector.**Input:** $d_{1}$, $SM_{1}$**Output:** *suspectlist*[]1.  Node *q*: /*monitoring node*/2.  Task 1:3.    **if** p∉suspectlist[] and did not receive heartbeat within freshpoint τ
4.      **then** add *p* to *suspectlist*[];5.    **end if**6.  Task 2:7.    upon receiving heartbeat message $m_{j}$ from *p*;8.      **if** $j>sn_{l}$
9.        **if** p∈suspectlist[]
10.         **then** remove *p* from *suspectlist*[];11.           ε=ε+1;12.         **else**
13.           ε=1;14.         **end if**15.         $d_{j}=\left(T_{now}-T_{pre}\right)/N_{l}$;16.         $EA_{j+1}=ID_{j}\cdot \eta_{j}+d_{j}$;17.         ${\hat{d}}_{j+1}=k*{\hat{d}}_{j}+(1-k)*d_{j}$;18.         $SM_{j+1}=\varepsilon \left|{\hat{d}}_{j+1}-d_{j}\right|$;19.         $\tau_{j+1}=EA_{j+1}+SM_{j+1}$;20.     **end if**21. Node *p*: /*detected node*/22.  **for** all i>1 **do**23.    at time $\eta_{i}$: (the *i*-th detection period);24.    send heartbeat message $m_{i}$ to node *q*;25.    $\eta_{i+1}\leftarrow \mathrm{Algorithm}\;1$;26.  **end for**

Sensor *p* as the monitored sensor sends heartbeat messages to sensor *q* every interval $\eta_{i}(i>0)$. Sensor *q*, as the detecting sensor, executes two tasks. One task will add sensor *p* into the suspect list when no heartbeat message from sensor *p* is received within the last freshpoint. The other task is responsible for computing the freshpoint based on the heartbeat message just received. After sensor *q* receives the heartbeat message, it can compute the communication delay and the safety margin of the next heartbeat message.

## 5. Evaluation and Performance

We conducted extensive simulations using actual data to evaluate the performance of our proposed failure detector and compared it with three other existing failure detectors. To improve the correctness of the experiment, we used the same method as that in paper [24], which applied the same data to replay different failure detectors and then computed the QoS metrics. This ensured that the comparative experiments were achieved in the same network condition.

### 5.1. Data Processing

Our experiments involved two nodes, one which represented the detecting node and the other that represented the monitored node. There was a communication channel between the nodes through a WiFi (802.11 g) network. One node as the monitored node was responsible for sending heartbeat messages, while the other node as the detecting node was responsible for receiving the heartbeat messages. Neither node failed during the experiment. The detecting node was equipped with a 900 MHz ARM Cortex-A7 processor, 1 G RAM, and a CentOS 6.5 operating system (Premier Farnell/Leeds). During the 3 h that the experiment lasted, heartbeat messages were generated at a target rate of one heartbeat every 100 ms. All heartbeat messages were transmitted using the UDP/IP protocol. In total, 88,011 heartbeat messages were sent, among which 87,800 were received (about 0.24% of message loss).

The distribution of arrival time of the heartbeat messages is shown in Figure 4a. From the figures, we can see that the arrival time of the heartbeat messages is concentrated around 100 ms and the heartbeat messages near 100 ms account for 92% of the total. Therefore, it is suitable to use the arrival time of the last heartbeat message to predict the arrival time of the next heartbeat message. Next, we selected the arrival time of heartbeat messages in three periods for observation (as shown in Figure 4b–d). The three periods represent the early, middle, and late stages of the experiment. From Figure 4b,c, we can observe that the arrival time of the heartbeat messages is concentrated around 100 ms. Additionally, the probabilities that the adjacent heartbeat messages have the same delay are 78.6% and 80.6%, respectively, in the early and middle stages of the experiment. From Figure 4d, it can be seen that the arrival time of the heartbeat messages is scattered. This may be caused by the dynamic network conditions; however, the probability that adjacent heartbeat messages have the same delay is 76.6%.

### 5.2. Discussions on Parameters

How the value of the timeout is set directly affects the performance of failure detection. A large timeout means a longer detection time when an actual node failure occurs. This will result in a possible drop in detection speed. On the other hand, a smaller timeout may cause a decrease in detection accuracy. In our failure detector, the value of the timeout was determined by the delay of heartbeat message and safety margin. There were two tuning parameters, *k* and ε, to compute the delay of heartbeat message and safety margin. As fine-grained *k* values can affect the performance of a failure detector, in our simulations, we computed the timeout through a series of *k* values, i.e., *k* = 0.1, 0.25, 0.5, and 0.75. ε was used to adjust the safety margin as another tuning parameter. In our simulations, we selected ε = 1, 1.5, and 2 to obtain the best failure detector performance. In practice, the optimal values of *k* and ε can be obtained via similar simulations or experiments.

In 2W-FD, there is a common tuning parameter called the safety margin, SM, in which users can obtain different detection times by setting different safety margins in their experiment. In accrual failure detectors, the tuning parameter is the threshold of φ-FD and ED-FD. The parameters of the algorithms are configured as follows: SM∈[0,1000]; for φ-FD, the parameters are set the same as those in [7,9]: Φ∈[0.5,16]; and $E_{d}\in [10^{-4},10]$ for ED-FD, as in [24]. Sliding window sizes are set as: 2W-FD: n_1_ = 1000 and n_2_ = 1. The algorithm can present the best performance of failure detection when it chooses these values compared to the bigger sliding window size.

The φ-FD and the ED-FD sizes are set at: n = 1000. These failure detectors have a better failure detection performance when they use the large window sizes [27]. Moreover, these failure detectors obtain minor improvements when the sliding window size exceeds n = 1000 in experiments. The same results are also mentioned in other papers [28]. Finally, the above parameter settings are the specific settings in their experiments.

### 5.3. Comparison of Failure Detection Metrics

The experimental results of the mistake rate vs. detection time are shown in Figure 5. The x-coordinate is used to indicate the detection time, and the y-coordinate is used to indicate the mistake rate. From Figure 5, we can see that the mistake rate of all failure detectors decreases with an increase in detection time. However, our failure detector had a lower mistake rate than other failure detectors when they had the same detection time. This improvement is because our failure detector can catch most late heartbeat messages by freshpoint under the same network conditions. When $T_{d}<0.29s$, the mistake rate of our failure detector is similar to 2W-FD and ED-FD. This shows that our failure detector can ensure detection accuracy during rapid detection. When $0.29s<T_{d}<0.34s$, the mistake rate of our failure detector had an obvious decrease compared with other failure detectors. This is because the calculation approach of the freshpoint can quickly adjust, so our failure detector adapts to the various network conditions better than other failure detectors.

The experimental results of query accuracy probability vs. detection time are shown in Figure 6. The x-coordinate is used to indicate the detection time, and the y-coordinate is used to indicate the query accuracy probability. The query accuracy probability of all failure detectors shows a consistency with the increase in detection time. When the detection time increases, the query accuracy probability of all failure detectors also increases. When $0.29s<T_{d}<0.34s$, the query accuracy probability of our failure detector had an obvious improvement compared with other failure detectors. This result is consistent with the measurement of mistake rate.

Figure 7 depicts the relative overhead comparison of two failure detectors. The 2W-FD represents the failure detector with a fixed detection period, while the LP-FD is the failure detector with a variable detection period. We observed that the 2W-FD with a fixed detection period introduced more traffic than our failure detector with a variable detection period in the early experiment (experiment time was less than 1.5 h). As time increases, the reliability of the sensor node decreases and the detection period becomes smaller; thus, the overhead of our failure detector continued to increase until it was the same as the failure detector with a fixed detection period.

### 5.4. Comparison of Battery Consumption

Sensors as static devices collect and transfer data to the sink node periodically. In addition, sensors can be used as relay nodes to forward data to other sensors [29,30]. From the above description, we employed two nodes to simulate the working environment of the IoT. Both the nodes were connected by a wireless link. Each node was not only a monitored node, but also a detecting node with a failure detector. There were multiple processes responsible for sending heartbeat messages or determining the state of the other node, respectively, run on each node. The nodes were equipped with an 800 mAh battery. In every experiment, different failure detectors were deployed on the nodes, then the running time of the nodes was measured.

These accrual failure detectors (φ-FD and ED-FD) and the 2W-FD generated more communication overhead than our failure detector at the same time. In addition, these failure detectors needed more calculation and storage, including the calculation of detector parameters and the storage of recent heartbeat messages in each detection period. To analyze the battery consumption of the failure detector, several sliding window size settings were selected (from n = 100 to 10,000), then a detailed comparison was made. For experimental reliability, each experiment of different failure detectors was carried out five times under the same environment and parameters. Finally, the running time of the node with different failure detectors was recorded. The experimental results are shown in Figure 8.

Figure 8 shows that the node without a failure detector deployed had the longest running time. 

We can see that the running time of the node without any failure detector is the longest. It improved by 10% compared to the node with the LP-FD. Among the nodes with a failure detector, the one that deployed φ-FD had the shortest running time, and the decrease is obvious with the increase in sliding window size. This may be because the battery consumption of the node is exacerbated when more heartbeat messages are sent, and a lot of calculations are done to ascertain the parameters of the normal distribution model. The fixed detection period introduces more heartbeat messages. While the node with the LP-FD had the longest running time. From the Figure 8, it can be seen that the improvement was up to 18% more than the φ-FD when the sliding window size was 10,000. In addition, the LP-FD did not need to maintain the sliding window and was thus unaffected by them. The nodes only maintained five connections in the experiment. In the actual IoT, each node needs to connect to many neighbor nodes to ensure the connectivity of systems. Therefore, the fact that the node that deployed the LP-FD could save battery consumption is more significant in real systems.

## 6. Conclusions

In this paper, we introduced our failure detector for environmental monitoring based on the IoT, namely the LP-FD. This failure detector possesses the capabilities to achieve sensor failure detection in a timely and accurate way. In order to save battery consumption and detection overhead, we computed the variable detection period by using the reliability function of the Weibull distribution. Moreover, our failure detector used both the prediction method of the last heartbeat message and the dynamic safety margin to ensure the accuracy of failure detection. According to the experimental results, we found that the LP-FD has a better detection speed, accuracy, overhead, and battery consumption than traditional failure detectors. Therefore, the LP-FD is suitable to provide failure detection services in the IoT.

## Figures and Tables

**Figure 1 sensors-21-06489-f001:**
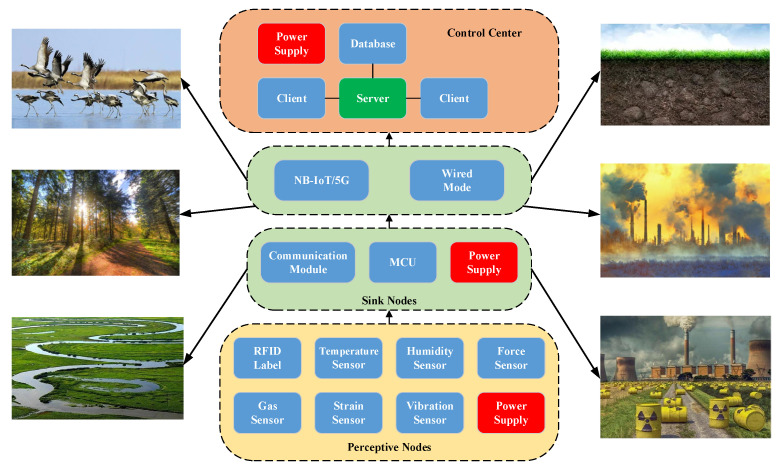
Structure of environmental monitoring based on the Internet of Things (IoT).

**Figure 2 sensors-21-06489-f002:**
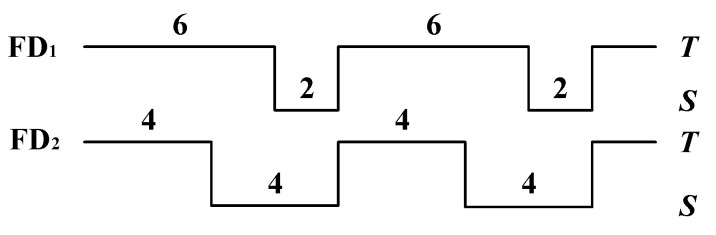
Query accuracy probability and mistake rate.

**Figure 3 sensors-21-06489-f003:**
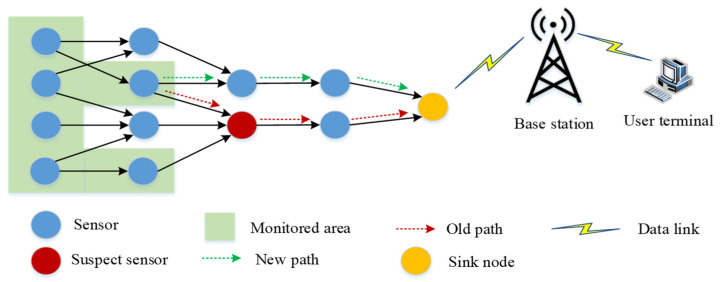
Example of sensor failure.

**Figure 4 sensors-21-06489-f004:**
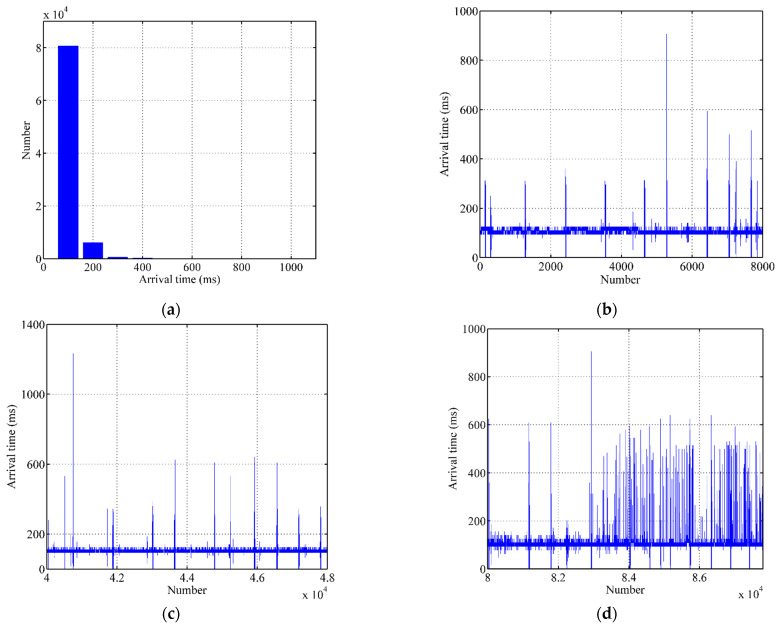
Analysis of actual data. (**a**) Distribution of arrival time, (**b**)Early stage of arrival time, (**c**) Middle stage of arrival time, (**d**) Late stage of arrival time.

**Figure 5 sensors-21-06489-f005:**
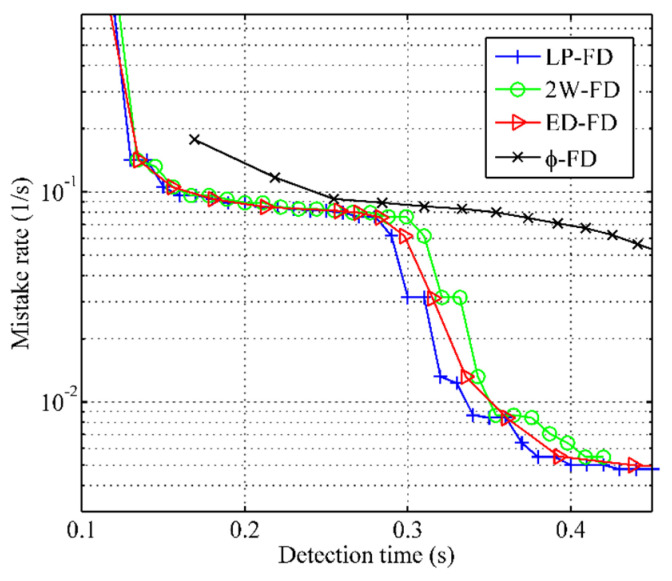
Mistake rate vs. detection time.

**Figure 6 sensors-21-06489-f006:**
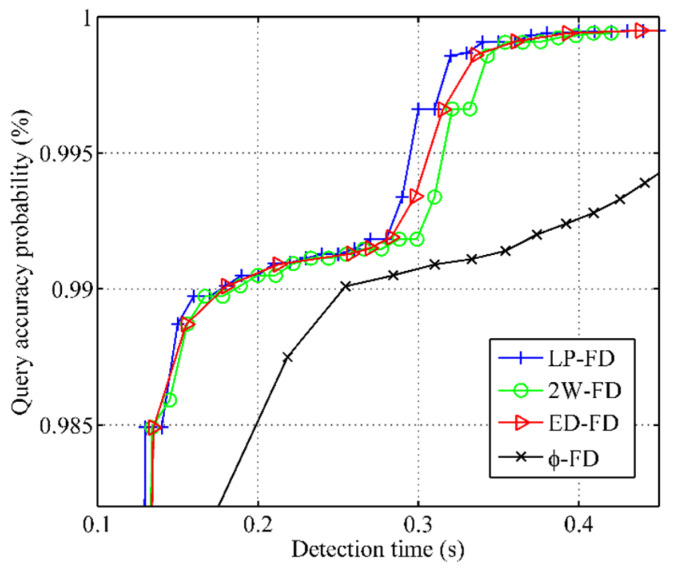
Query accuracy probability vs. detection time.

**Figure 7 sensors-21-06489-f007:**
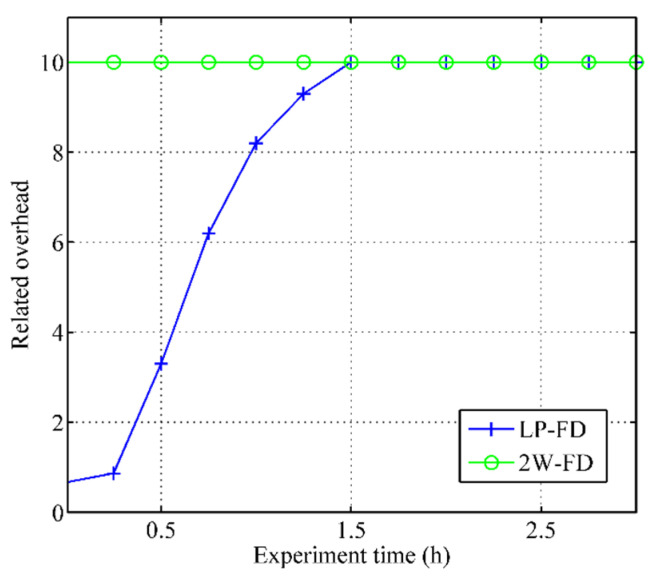
Comparison of related overhead.

**Figure 8 sensors-21-06489-f008:**
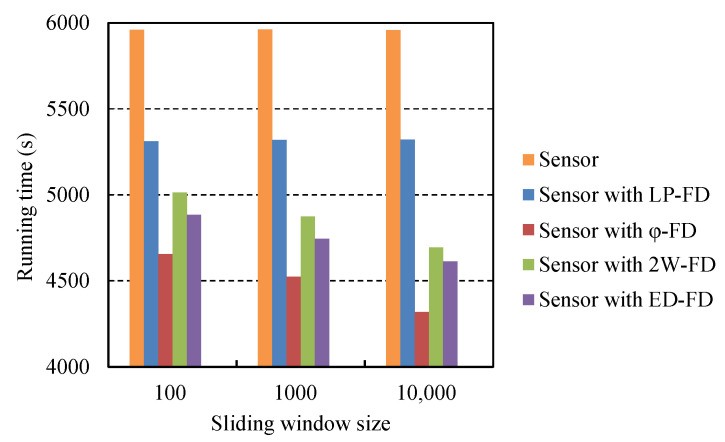
Comparison of battery consumption.

**Table 1 sensors-21-06489-t001:** Definition of parameters.

Symbol	Definition
η	detection period
R(t)	reliability function of Weibull distribution
α	scale parameter of Weibull distribution
β	shape parameter of Weibull distribution
Δt	time difference between a certain time and calculated time based on reliability
$\eta_{\min}$	preset minimum detection period
*EA*	expected time arrival of next heartbeat message
τ	freshpoint of heartbeat message
*SM*	safety margin
$\hat{d}$	predictive heartbeat message delay
*d*	heartbeat message delay
$T_{now}$	arrival time of new heartbeat message
$N_{l}$	number of lost heartbeat messages
$ID_{i}$	sequence number of heartbeat messages

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
