# Peer review of "Low-Power Failure Detection for Environmental Monitoring Based on IoT"

_sensors, 2021, doi:10.3390/s21196489_

Round 1

Reviewer 1 Report

This paper addresses the problem of failure detection when remote nodes do not have sufficient power supply. Especially, it proposes a LP-FD using the variable detection period.

First of all, one big problem of this manuscript is that it seriously needs to be proofread by a native speaker because there are lots of fundamental grammatical errors.

Although the idea of various detection period for the reduction of required battery consumption is quite simple and straightforward, the explanation of the proposed algorithms and equations is not sufficient for readers to understand. To make the manuscript more descriptively clear and scientifically sound, I have some questions and comments.

  1. (Page 4) There are missing definitions of notations and terminologies. In equation (1), what is k ? For readers who are not familiar with the failure detection area, it would be better to give definition of fundamental terminologies such as fresh point, safety margin, and so on.
  2. (Page 7, line 256) what do authors mean by n ?
  3. (Page 9, algorithm 2, line 8) what is snl ?
  4. (Page 9, algorithm 2, line 11) is there any specific reason to update ε to ε+1 ? For example, what if the value is updated to ε+2 or ε+3 ?
  5. (page 10, Fig 4(a) ) As the experiments are conducted in LAN, the graph does not show the arrival time in WAN that seems to be more realistic in environmental monitoring?
  6. (page 10, Fig 4(d) ) At the last stage of the experiment, it seems that the relatively high arrival time was measured more frequently than (b) and (c). I am curious about the reason why this situation happened.
  7. (page 11, line 354-355) What are the optimal values of k and ε ?
  8. (page 13, Fig 7) What is the unit of y-axis? What about the overhead of the other approaches Fig 5,6, and 8 ?

Reviewer 2 Report

  1. The authors propose a low power failure detector, which can provide the acceptable failure detection service and save the battery consumption of nodes.
  2. Their simulation results show that LP-FD can provide better detection speed, accuracy, overhead and battery consumption than other failure detection algorithms.
  3. In the figure 2, the query accuracy probability and mistake rate can be demonstrated in detail.
  4. In the figure 3, example sensor failure can be demonstrated in detail.
  5. Please revise and enhance the English writing to increase readability.

Round 2

Reviewer 1 Report

I think that the revised manuscript satisfies previous review comments and questions.

Reviewer 2 Report

no comment.